# SILG: The Multi-environment Symbolic Interactive Language Grounding Benchmark

Victor Zhong[1,3], Austin W. Hanjie[2], Sida I. Wang[3], Karthik Narasimhan[2] and Luke Zettlemoyer[1,3]

[1]Department of Computer Science, University of Washington
[2]Department of Computer Science, Princeton University
[3]Facebook AI Research

## Abstract

Existing work in language grounding typically study single environments. How do we build unified models that apply across multiple environments? We propose the multi-environment Symbolic Interactive Language Grounding benchmark (SILG), which unifies a collection of diverse grounded language learning environments under a common interface. SILG consists of grid-world environments that require generalization to new dynamics, entities, and partially observed worlds (RTFM, Messenger, NetHack), as well as symbolic counterparts of visual worlds that require interpreting rich natural language with respect to complex scenes (ALFWorld, Touchdown). Together, these environments provide diverse grounding challenges in richness of observation space, action space, language specification, and plan complexity. In addition, we propose the first shared model architecture for RL on these environments, and evaluate recent advances such as egocentric local convolution, recurrent state-tracking, entity-centric attention, and pretrained LM using SILG. Our shared architecture achieves comparable performance to environment-specific architectures. Moreover, we find that many recent modelling advances do not result in significant gains on environments other than the one they were designed for. This highlights the need for a multi-environment benchmark. Finally, the best models significantly underperform humans on SILG, which suggests ample room for future work. We hope SILG enables the community to quickly identify new methodologies for language grounding that generalize to a diverse set of environments and their associated challenges.

## 1 Introduction

An ideal language-conditioned agent should interpret language in diverse environments with varying observation space, action space, language, and plan complexity. However, existing language-grounding literature typically focuses on single environments, and proposes methodological contributions specific to those environments [35, 51]. In order to determine which contributions are environment-specific and which apply across multiple environments, it is critical to develop universal models that can be easily evaluated in many different settings.

To facilitate this research, we present the multi-environment Symbolic Interactive Language Grounding Benchmark (SILG). We focus on symbolic environments with semantic symbols instead of raw visual observations for efficiency, interpretability, and emphasis on abstractions over perception. SILG consists of diverse environments including grid-worlds RTFM [58], Messenger [22], and NetHack [34], which require generalization to new dynamics (i.e. how entities behave), entity references, and partially observed worlds. SILG also contains symbolic counterparts of visual grounding

---

Corresponding author Victor Zhong `vzhong@cs.washington.edu`

35th Conference on Neural Information Processing Systems (NeurIPS 2021).

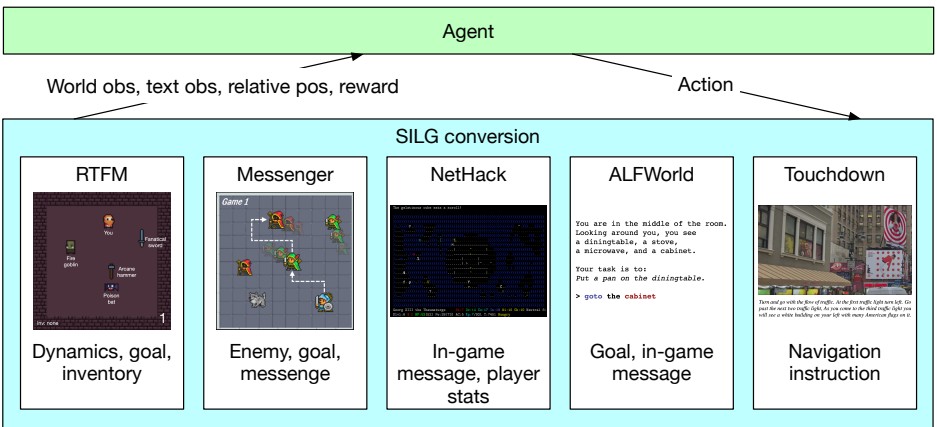

Figure 1: Environments included in SILG. The world observations and text fields are shown for each environment. Detailed examples are in Appendix F.

environments ALFRED [48] and Touchdown [9] , which require interpreting rich natural language in complex scenes. For the former, we use its textual variant ALFWorld [49]. For the latter, we create SymTD by applying object segmentation to Touchdown panoramas. Despite significant implementation differences, we unify these environments under a common interface in SILG, so that one can easily develop and evaluate language grounded RL methods across all of these challenges.

SILG environments present a variety of unique grounding challenges in the richness of the observation space, action space, language specification, and plan complexity. We quantify these challenges and additionally analyze the success rate and lengths of expert playthroughs. For visual grounding environments, we show symbolic variants (ALFWorld and SymTD) facilitate faster learning and result in policies that transfer to their visual counterparts. While a unified model may not outperform specialized models engineered for specific environments, it can be helpful to understand whether particular modelling innovations are environment specific or more general techniques. Furthermore, while the challenges in each environment are very different, we want to encourage the development of unified architectures and approaches that can scale across many language grounding tasks.

In addition to SILG, we propose the Symbolic Interactive Reader (SIR), the first shared model architecture for these environments. We combine SIR with several recent advances in language-conditioned RL, including $FiLM^2$ [58], egocentric local convolution [27], recurrent state-tracking [34], entity-centric attention [22], and large pretrained LMs [28]. On most environments, SIR achieves comparable performance to methods designed specifically for single environments. In addition, we find that many recent advances do not result in significant gains on environments other than the one they were designed for. This highlights the need for a multi-environment benchmark. Finally, the best models significantly underperform humans on SILG (10-85% depending on environment), which suggests ample room for modelling improvements that generalize across environments.

In summary, we (1) combine five language-grounding environments under the same interface to evaluate language grounded RL methods across diverse grounding challenges, (2) present the first shared model architecture for these environments, and (3) analyze recent modelling contributions across these environments. We hope SILG enables the community to quickly identify new models and learning algorithms that generalize to a diverse set of environments and their associated challenges. The code for SILG is available at `https://github.com/vzhong/silg`.

## 2 SILG Environments

SILG contains five language-grounding environments including both grid-worlds (RTFM, Messenger, SILGNetHack) and symbolic counterparts of 3D-visual worlds (ALFWorld, SymTD). While all involve agents situated in interactive worlds, each presents unique challenges in richness of observation space, action space, language specification, and plan complexity. Table 1 quantifies their theoretical complexity along these dimensions as well as empirical complexity using expert playthroughs.[1]

---

[1]For each environment, an expert plays as many episodes as necessary to learn about the game. We then record the playthroughs to compute the empirical win rate and trajectory length. More details in Appendix F.

Table 1: SILG statistics. "dynamics" are high level rules dictating behaviour of entities. "Ref hops" are number of intra-text references the agent must resolve to determine correct course of action. Messenger and SymTD text are human-written instead of procedurally generated. Distinctive properties are **bold**.

| | RTFM | Messenger | SILGNetHack | ALFWorld | SymTD |
|---|---|---|---|---|---|
| **Action space** | 5 fixed | 5 fixed | 23 fixed | **50+ choices** | 1-5 choices |
| **State space** | 6 × 6 grid
5 entities | 10 × 10 grid
14 entities | 21 × 79
**partial obs** | 102 nodes
191 entities | **29.6k complex**
**panoramas** |
| **Mean text len** | 31 words | 30 words | 9 words | **100 words** | **90 words** |
| **Vocab size** | 262 words | 595 words | ∼100 words | 1237 words | **4999 words** |
| **Generalization** | **new dynamics** | **new dynamics** | new layouts | new instr
new layouts | new instr |
| **Ref hops** | **6 hops** | 3 hops | 1 hop | ∼4 hops | **∼7 hops** |
| **Human win %** | 100% | 100% | 78.1% | 100% new instr
100% +layouts | 61.5% |
| **Human # steps** | 6.0 steps | 2.2 steps | 34.4 steps | 7.8 steps new instr
9.6 steps +layouts | 33.6 steps |
| **Env FPS** | 240 | 1627 | 439 | 7 | 779 |
| **Key challenge** | multi-step
reasoning | adversarial
generalization | partial obs | large
action space | complex
language |

The goal of SILG is to provide a simple-to-use benchmark that allows researchers to quickly evaluate methods across all of these environments as well as their respective challenges. We thus combine these environments under a unified interface built on top of OpenAI Gym [7]. In each environment instance, the agent observes text inputs as well as world observations. For grid worlds such as RTFM, Messenger, and SILGNetHack, the agent receives a 2-D bird's-eye-view symbolic grid as observations. For visually inspired environments such as ALFWorld and SymTD, the agent receives a symbolic egocentric view of the present scene. Figure 2 shows how SILG environments are rendered to players via the `play` utility. In the rest of this section, we describe each SILG environment in detail. Appendix B shows how to use SILG in Python. Appendix G shows licensing for SILG environments.

**Selection criteria** We select interactive environments that span the challenges presented in Table 1, are easily converted to symbolic representations, and avoid the use of additional simulators (e.g. Matterport3D [1]). While visual perception is clearly important for language grounding [19], we focus on the unique challenges of symbolic environments such as multi-hop reasoning and generalization to rich sets of procedurally generated dynamics. We leave the challenge of developing a visually rich multi-environment grounding benchmark to future work. Due to the lack of gold trajectories in many of the selected environments, we do not support imitation learning (IL) in this version of SILG.

**RTFM** RTFM [58] is a grid-world environment where an agent interprets text to acquire the correct items to fight the correct monsters. A key challenge in RTFM is multi-modal multi-step reasoning (at least 6 steps) combining world observations with texts associated with multiple entities. Given a team to beat, the agent must identify which monster is on the team, then identify the item descriptor that would beat the monster descriptor. Finally, the agent must acquire the item with the correct descriptor and engage the correct monster to win. RTFM evaluation is on games with unseen rules, forcing agents to make novel reasoning steps to generalize successfully. At each step, the agent receives a symbolic grid containing names of entities present, as well as texts indicating the high level rules, the agent inventory, and the goal of the particular game instance. We include all 4 RTFM curriculum stages, but only show results for the first stage in this preliminary study.

**Messenger** Messenger [22], is a grid environment where the agent must acquire a message and deliver it to the goal while avoiding an enemy after extracting entity-role assignments from a text manual. A key challenge in Messenger is the adversarial train-evaluation split without prior entity-text grounding. There is no overlap in entity-role assignments between training and evaluation, forcing agents to make compositional entity-role generalizations. At each step, the agent receives a symbolic grid containing symbol IDs of entities present, as well as texts indicating roles of each entity. The

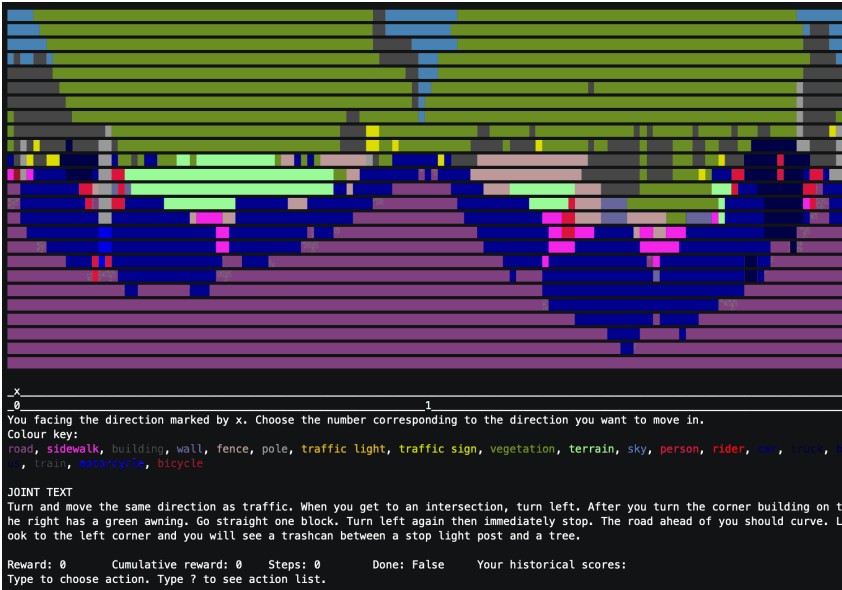

Figure 2: The SILG `play` utility (shown here for SymTD) enables human playthrough as well as visualizing what input the model observes. Because all environments are symbolic, the `play` utility works in console (e.g. via `ssh`, `tmux`) without need for X-forwarding.

entities are referred to in text by many names, which have no lexical overlap with their symbol ID. That is, the text "dog" in the text for example is the non-textual symbol 2 in the observation and the association between entities and references must be learned via interaction. We include all 3 Messenger curriculum stages, but only show results for the first stage in this preliminary study.

**SILGNetHack** NetHack is a a complex rogue-like game from the NetHack learning environment [34]. In SILGNethack, we combine 3 tasks (`Score`, `Gold`, and `Scout`) and specify the task to complete for each episode via a text prompt. SILGNetHack is challenging due to its large state space and partial observability. The agent may descend multiple floors and sections of each floor may be obscured until exploration by the agent. Because of the different score distributions of each task, we mark a trajectory as successful if it exceeds a task-specific score threshold determined from human playthroughs. We evaluate agents on previously unseen map layouts that are procedurally generated with new seeds disjoint from the ones used during training. More information about the SILG multi-task SILGNetHack is in Appendix D. At each step, the agent receives a symbolic grid containing symbol IDs of entities present, as well texts denoting the goal, agent stats, and feedback from the environment after the agent's last action. SILGNetHack vocabulary is technically infinite because players can arbitrarily name things, however in our expert playthroughs of SILG SILGNetHack, we observe just over 100 unique words. Human experts win just under 80% of games with an average of 34 steps, which demonstrates the challenge of SILGNetHack. All failures can be attributed to hitting the step limit before acquiring the necessary win conditions.

**ALFWORLD (text ALFRED)** In ALFWorld, an agent navigates and manipulates objects inside a 3D kitchen [49]. Its large text action space, with more than 50 valid actions (given by the game engine) for most scenes is a key challenge. Unlike its visual counterpart ALFRED [48] where the agent observes 3-D images of the kitchen, in ALFWorld the agent must rely on language descriptions of the kitchen. Goals are provided in human written language (e.g. put a clean sponge on the metal rack). The language in ALFWORLD is not complex, but are 100 words on average due to a large number of items in a single scene. Following recent work [49], we evaluate on both unseen instructions (new instr) and unseen room layouts (new layouts). At each step, the agent receives the goal text and a list of items present in the room (e.g. "cup 1", "bottle 2"). We concatenate the names of these items into a symbolic world observation grid , each entry containing the name of one item. The agent then selects from plausible commands given what is present in the scene.

**SILGTouchdown (SymTD, VisTD)** In Touchdown, the agent navigates through Google Street View panoramas according to long compositional instructions that tests spatial reasoning [9, 37, 38]. A key challenge is the rich human-written navigation instructions that describe photorealistic images. Touchdown's long human-written instructions contain many intra-text reference hops, which we

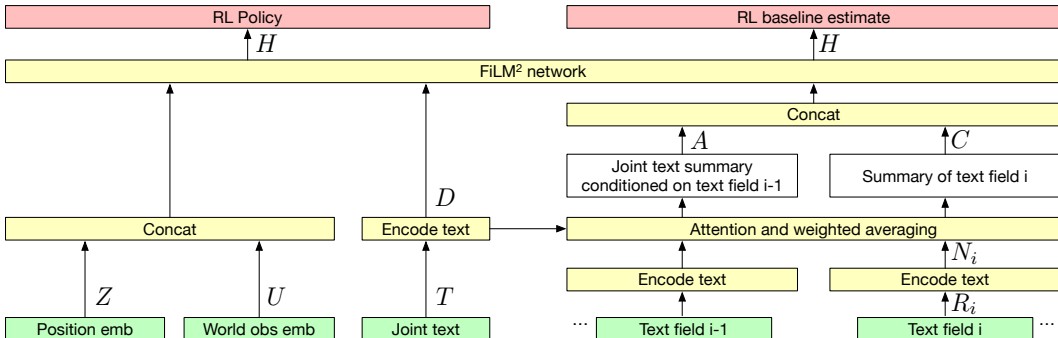

Figure 3: The Symbolic Interactive Reader (SIR) baseline. Inputs are green, intermediate results white, outputs red, and model components yellow. Details about the $\text{FiLM}^2$ layer is in Appendix C.

approximate as the number of sentences plus the number of sequential connectors such as "then". We convert Touchdown to a symbolic environment by segmentating its panoramas into semantic grids. In each step, the agent observes the instruction text and a grid of discretized segmentation class IDs corresponding to the current panorama. It then chooses among a list of radial directions to proceed to the next panorama. The agent wins if it passes the goal location. We use the same train-test split as the original Touchdown environment, which features unseen navigation texts.

We show that our symbolic Touchdown (SymTD) facilitates faster learning compared to learning in its visual equivalent (VisTD). Human performance demonstrates some limitations of SymTD, with an expert win rate just over 60%. This may be due to the symbolic representations removing information referenced by the instructions such as color, or because the segmented features are visually disparate from real-world views [17]. We also include manual stop variants of SymTD and VisTD, which are functionally equivalent to the original Touchdown. Appendix E details these variants, SymTD/VisTD creation as well as discussions on human performance. Compared to prior work on Touchdown and ALFWorld, we train using RL without supervised trajectories as opposed to imitation learning.

## 3   The Symbolic Interactive Reader Baseline Model

Figure 3 shows the SIR baseline for the SILG benchmark. To the best of our knowledge, this is the first shared model architecture for RTFM, Messenger, NetHack, ALFWorld, and Touchdown. Consider an agent situated in an arbitrary SILG environment. At each time step $t$, the model receives from the environment the following inputs (precise inputs for each environments are shown in Appendix F).

- **World observations** $X \in \mathbb{R}^{h \times w \times k}$ where $h$ and $w$ are the height and width of the observation and each element corresponds to the $k$-word symbol ID(s) of its content.

- **Joint text** $T \in \mathbb{R}^l$ of $l$ tokens of the text to attend over.

- **Text fields** $R \in \mathbb{R}^{n \times m}$ where the $i$th row contains the $i$th of $n$ environment text field such as agent inventory or environment feedback. $m$ is the max token count of these texts.

- **Relative position** $Z \in \mathbb{R}^{h \times w \times 2}$ cell-wise feature that denotes the position of each cell relative to the player agent in the $x$ and $y$ directions.

As a policy learner, the model must output a distribution $Y$ over the action space. We additionally output a baseline estimate of the value function to stabilize policy learning [18]. Let $d$ and $r$ denote embedding and bidirectional LSTM sizes. We first sum embeddings for each cell in the world observation to obtain world representation $U = \text{sum}\,(\text{emb}\,(X)) \in \mathbb{R}^{h \times w \times d}$. Next, we encode the $i$th text field $R_i$ and the joint text $T$ using an bidirectional LSTMs [30].

$$N_i = \text{BiLSTM}_N\,(\text{emb}\,(R_i)) \in \mathbb{R}^{m \times r} \tag{1}$$

$$D = \text{BiLSTM}_D\,(\text{emb}\,(T)) \in \mathbb{R}^{l \times r} \tag{2}$$

We then compute weighted average over text fields $\tilde{C}_i$ and attention $\tilde{A}_i$ over the joint text.

$$\tilde{C}_i = \text{weightave}_i(N_i) = \sum_j \text{softmax}(\text{linear}_i(N_i))_j N_{ij} \in \mathbb{R}^r \qquad (3)$$

$$\tilde{A}_i = \text{attend}\left(D, \tilde{C}_i\right) = \sum_j \text{softmax}\left(D\tilde{C}_i\right)_j D_j \in \mathbb{R}^r \qquad (4)$$

We compress $\tilde{C} \in \mathbb{R}^{n \times r}$ and $\tilde{A} \in \mathbb{R}^{n \times r}$ again to support any number of text fields.

$$C = \text{weightave}_C\left(\tilde{C}\right) \in \mathbb{R}^r \qquad (5) \qquad\qquad A = \text{weightave}_A\left(\tilde{A}\right) \in \mathbb{R}^r \qquad (6)$$

We now have representations for world observations $U$, text fields $C$, and joint text conditioned on text fields $A$. We apply successive FiLM$^2$ layers to build multiple levels of codependent representations between texts and world observations to model multiple cross-modal reasoning steps [58]. To support arbitrary number of text fields, we modify the text input of the $i$th FiLM$^2$ layer to be the concatenation of the text fields $C$, attention over joint text conditioned on text fields $A$, and attention over joint text conditioned on the visual summary of the last FiLM$^2$ layer $s^{(i-1)}$.

$$V^{(i)}, s^{(i)} = \text{FiLM}^2\left(\left[V^{(i-1)}; Z\right], \left[C, A, \text{attend}\left(D, s^{(i-1)}\right)\right]\right) \qquad (7)$$

We use the definition of FiLM$^2$(visuals, texts) from Zhong et al. [58] and summarize its intuition and computation in Appendix C. We define $V^{(1)}$ and $s^{(1)}$ to be the initial world observation $U$ and its spacial max-pooling. Finally, we use a multi-layer perceptron to build a fixed-size codependent representation of the inputs based on the last FiLM$^2$ layer's output $H = \tanh\left(\text{linear}_4\left(\text{flatten}(V^{(\text{last})})\right)\right)$, which is used to compute the baseline estimate of the value function $B = \text{MLP}_B(H)$ and the policy $Y(H)$ expressed as a distribution over actions. While the core architecture of SIR is identical for all environments, a different policy module $Y$ is necessary for different types of action spaces.

**Fixed sized action space (RTFM, Messenger, SILGNetHack)** We simply apply a multilayer perceptron to the final representation $Y = \text{MLP}_Y(H)$.

**Multiple-choice text action space (ALFWorld)** Let $Q_j$ denote tokens for the $j$th choice (e.g. pick up the mug), which we encode a bidrectional LSTM $G_j = \text{BiLSTM}_G(\text{emb}(Q_j))$. We then attend over this text using the final representation $H$ to score for $j$th choice $Y_j = \text{linear}_4(\text{attend}(G_j, H))$.

**Multiple-choice navigation action space (SILGTouchdown)** Let $j$ denote the index of the world representation corresponding to a movement direction. For example, for a world observation width of 100, the index corresponding to advancing in the 30 degrees direction is $\frac{30*100}{360} \approx 8$. We encode the navigation choice by selecting its corresponding world observation representation, then scoring it via dot product with the final output representation $Y_j = \text{linear}_5(U_j)^\intercal H$.

## 4 Experiments

**Setup** How well does a shared architecture do across all five SILG environments? To answer this, we train and evaluate SIR using Torchbeast [33], a distributed RL framework with importance weighted actor-learners based on IMPALA [18]. For each environment (separately), we train on training, do early stop on validation, and evaluate on test. NetHack does not distinguish between train and evaluation, hence we create our own splits by dividing the seed range (first 1 million seeds for training, second for validation, and third for test). We run 5 random seeds for each environment. The hyperparameter and compute resources are respectively shown in Appendix H and I. SILG.

**Results** Figures 4 through 8 show learning curves for each environment. Table 2 shows the test performance for the baseline model and the best model variant. Despite sharing the same core model architecture, SIR achieves reasonable performance across all environments except Messenger, where it overfits due to lack of pretrained LM and entity-centric attention. Nevertheless, the best performing model significantly trails human performance, indicating room for further improvement.

### 4.1 Analyses of recent grounded language RL modelling contributions

Next, we use SILG to evaluate recent modelling advances for language grounding across environments by adding them to the SIR baseline. These modelling enhancements were proposed for (and resulted in key gains on) the environments included in SILG. Namely, we analyze the effectiveness of recurrent state-tracking, entity-centric local convolution, entity-centric attention, and pretrained LMs.

Table 2: Success rate on test environments for SIR and its best variant. Standard deviation are in brackets. We early stop on validation and evaluate best checkpoint on test. For RTFM, Messenger, and SILGNetHack, we evaluate 100 episodes. For ALFWorld and Touchdown, we evaluate on initial states from each test episode. The variant with best performance across envs is +state. The SOTA for RTFM, Messenger, and ALFWorld are respectively from Zhong et al. [58], Hanjie et al. [22], and Shridhar et al. [49] (std was not reported in ALFWorld). $^\triangle$SOTA for ALFWorld relies on supervised trajectories and beam search, which SIR does not use. There are no previous results for multitask SILGNetHack and SymTD as they are introduced here. Though not comparable, the manual stop VisTD SOTA trained using imitation learning on supervised trajectories is 16.7% [56].

| Model | RTFM | Messenger | SILGNetHack | ALFWorld | | SymTD |
| --- | --- | --- | --- | --- | --- | --- |
| | | | | new inst | new inst+layouts | |
| Base | 88.8 (22.4) | 0 (0) | 23.8 (0.8) | 21.0 (1.5) | 16.0 (2.1) | 9.7 (1.3) |
| Best | +state | +all | +local conv | +state | +state | +state |
| | 99.2 (0.7) | 31 (2.6) | 25.4 (3.3) | 23.6 (2.8) | 16.6 (2.9) | 14.9 (1.8) |
| SOTA | 83 (21) | 85 (1.4) | N/A | $40^\triangle$ | $37^\triangle$ | N/A |
| Human | 100 | 100 | 78.1 | 100 | 100 | 61.5 |

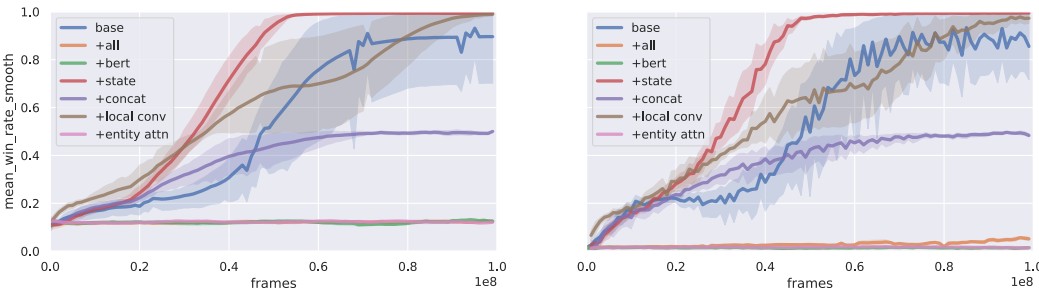

Figure 4: RTFM performance. Left: train envs, right: validation envs.

**Recurrent state tracking (`state`)** As in Küttler et al. [34], we augment the SIR baseline with a state-tracking LSTM by replacing the final $H$ with $H' = H + \text{LSTM}(H, S_{t-1})$, where $S_{t-1}$ is the previous LSTM state (summing LSTM output and $H$ outperforms replacing $H$ with LSTM output). State-tracking consistently improves convergence and generalization, even when the correct next step is fully determined by current world observations (e.g. RTFM). This may be because it helps prevent local minima that cause repetitive actions. The exception to this is Messenger, where state-tracking does not help generalize to the evaluation distribution.

**Entity-centric local convolution (`local conv`)** Hill et al. [27] proposed local convolution around the agent to obtain an egocentric view of world observations. While this helps generalize in SIL-GNetHack, it does not help significantly in other environments. One reason is that this provides redundant information as positional embeddings, which is already included in the base model and is a cheaper alternative to adding an additional egocentric convnet.

**Entity-centric attention (`entity attn`)** Hanjie et al. [22] propose replacing entity representations with attention over text specification, such that the world observations are forcibly composed using text representations. We add this by replacing world representation $U$ with entity attention over text fields $R$ as described in Hanjie et al. [22]. This constraint causes underfitting of SIR on most environments. Since the entity representation is built entirely using the text, when there is incomplete entity information or it is difficult to extract the relevant information from the manual text this can be a handicap. However, for Messenger, entity-centric attention prevents overfitting.

**Pretrained language model (`bert`)** A natural question in language-grounding is how to leverage large, pretrained LMs [28]. We use a simple method to incorporate BERT [16] by replacing all text encoding with the summation of the original bidirectional LSTM encoding and BERT encoding. Due to the memory requirement of large pretrained LMs, we cannot fine-tune the LM during training, and thus keep the LM parameters fixed. Pretrained LMs (`bert` and `all`) help generalization in Messenger but does not improve performance on other environment in our experiments. For tasks such as RTFM

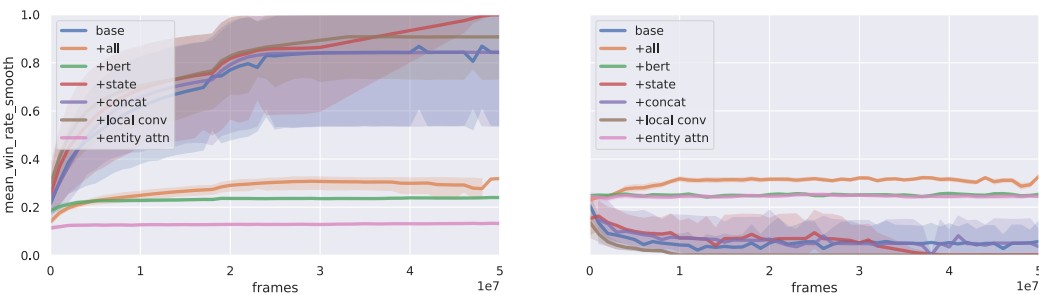

Figure 5: Messenger performance. Left: train envs, right: validation envs.

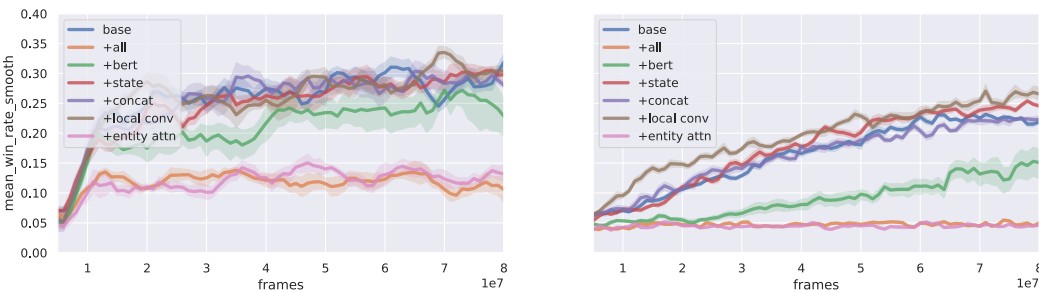

Figure 6: SILGNetHack performance. Left: train envs, right: validation envs.

and SILGNetHack, our use of a general-purpose LM may not be beneficial for the highly specific language used in those tasks (i.e. fantasy world with word like shaman, goblin, mage etc). We stress that this is a preliminary investigation into the use of LMs on these environments, and we encourage future research on how to effectively use pretrained LMs across environments using SILG.

## 4.2 Analyses of SILG environments

Finally, we examine performance of SIR and variants to analyze challenges presented by SILG.

**Generalization requirement of environments**  SILG's evaluation environments require different types of generalization. RTFM requires generalizing to new environment dynamics by referring between world observations and multiple texts; because SIR adopts FiLM$^2$ from Zhong et al. [58], it is able to achieve such generalization. Messenger requires compositional entity-role generalizations. That is, if an entity (e.g. dog) has a certain role (e.g. message holder) in training, such an entity-role assignment never appears in validation or test. SIR quickly overfits to to entity-role assumptions (e.g. dog as the message) in training suggesting the need for additional work on achieving this type of generalization using a joint model architecture. Combining pretrained LM with other enhancements (`+all`) results in generalization improvement, however the convergence remains very slow. This suggests that generalizing to new dynamics across environments without obvious lexical cues from the text remains a difficult challenge. SILGNetHack and ALFWorld require generalizing to new procedurally generated scenes, which SIR achieves. In the additional out-of-domain ALFWorld evaluation where the model must generalize to new layouts, state-tracking allows the model to generalize faster. Touchdown requires generalizing to new natural language instructions. Here, the baseline suffers from a large generalization gap. We hypothesize that more effective means of incorporating pretrained LMs is necessary to achieve this type of generalization.

**Necessity of separate text fields**  In `concat`, we concatenate text fields into a single string, which we encode using a bidirectional LSTM. In this case, both joint text $D$ and text field representations $N$ are set to this encoding. This degrades performance especially in RTFM, which shows that multi-hop references is more easily learned when the text fields are separated and modeled via structured attention. Note that this model variant is not shown for Touchdown because it only has one text field.

**Learning from symbolic vs visual world observations**  Table 3 shows that policies learned in the symbolic environment transfer to the 3-D environment. Using oracle and Masked-RCNN [24], the ALFWorld policy can be transferred by filling observation text templates using detected objects. Our result with oracle detector is in line with Shridhar et al. [49], though our performance is weaker

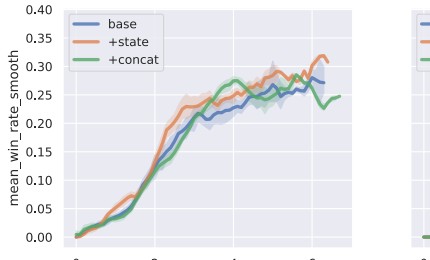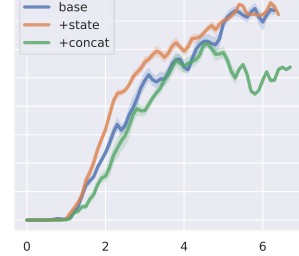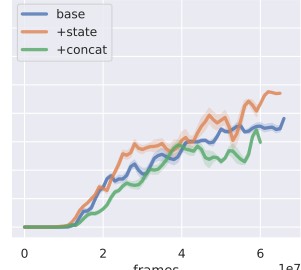

Figure 7: ALFWorld perfomance. Left: train envs, middle: new instruction validation envs, right: new instruction+new layouts validation envs. For efficiency we only evaluate on a subset (50 out of 140) of the validation environments for early stopping. We do train BERT variants here due to computational constraints. ALFWorld does not have entity IDs and no agent location, hence we do not show local convolution nor entity attention experiments.

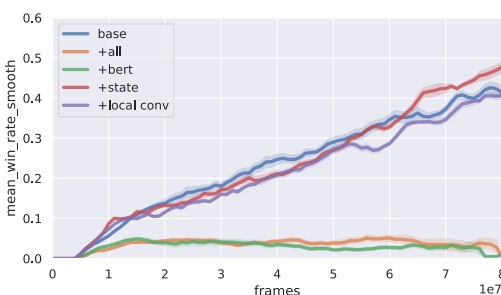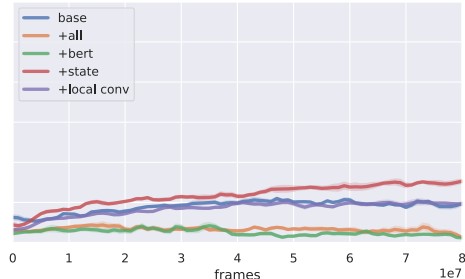

Figure 8: SymTD performance. Left: train envs, right: validation envs. Touchdown does not have entities, hence we do not show experiments for entity attention.

because we do not use annotated data nor DAgger [44]. As with prior results, transfer to visual worlds with new layouts remains very challenging [49]. Transfer using Masked-RCNN results in large drop in performance, nevertheless SILG allows perception, albeit an important challenge, to be factored out so that one can focus and quickly iterate on abstraction challenges. Table 3 also shows that models trained on SymTD outperform those trained on VisTD (where $U$ is 10-dim PCA features from a ResNet [23] panorama encoding) despite being faster (383 for SymTD vs. 344 frames per second for VisTD). That is, by applying segmentation to obtain SymTD, we are able to obtain a better policy than training directly with visual features using VisTD. The results from both ALFWorld and SymTD show that learning in faster symbolic environments such as SILG can transfer to their visual counterparts, and allows certain perception challenges to be factored out.

**Future work** We find that some of the most challenging aspects of situated interactive language grounding include (1) grounding text references to entities without lexical overlap, (2) choosing from large textual action spaces, and (3) interpreting complex natural language descriptions. On the methodology front, further work is needed to investigate how to effectively use pretrained LMs for language grounding. Moreover, apart from recurrent state tracking, the other model enhancements do not yield significant gains on environments other than the ones they were proposed for. These results highlight the need for modelling techniques that generalize across environments.

SIR suggests that with additional improvements, it may be possible to have a performant model with the same architecture (but trained independently) across environments. Future work may explore whether (1) a single model with the same parameters can accomplish all tasks, (2) a single model with pretraining can be quickly finetuned on each task, and (3) learning in one environment is transferable to another. We believe SILG is well-suited to help answer these questions. Furthermore, SILG is designed to be easily extensible, with opportunities to add additional environments in the future.

## 5  Related Work

**Benchmarks for NLP and RL.** NLP benchmarks helped the development of models that generalize across different tasks [53, 54]. Similar benchmarks have furthered research in RL [10, 12, 50]. SILG is the first benchmark for symbolic interactive language grounding with a diverse set of language and

Table 3: Transfer task success rate from symbolic to visual envs for baseline and its best variant. Standard deviation shown in brackets. For ALFRED, we give ALFWorld-trained models language templates filled with detected objects from vision using oracle and Masked-RCNN object detectors.

| Model | ALFWorld/ALFRED | | | | | | SymTD | VisTD |
|---|---|---|---|---|---|---|---|---|
| | new inst | | | new inst + layouts | | | to | |
| | text | oracle | m-rcnn | text | oracle | m-rcnn | VisTD | |
| Base | 21.0(1.5) | 11.2(3.3) | 3.0(1.7) | 16.0(2.1) | 0.7(0.4) | 0.3(0.2) | 9.7(1.3) | 4.3(4.0) |
| Best | +state | | | +state | | | +state | base |
| | 23.6(2.8) | 11.3(1.9) | 7.1(1.1) | 16.6(2.9) | 1.3(1.1) | 0.7(0.6) | 14.9(1.8) | 4.3(4.0) |

**RL challenges.** SILG evaluates generalization to new dynamics with (RTFM) and without lexical cues (Messenger) between text and entities, large partially observed worlds (SILGNetHack), large actions spaces (ALFWorld), and complex natural language instructions in rich visual scenes (SymTD). Finally, SILG provides a standard interface for symbolic interactive grounding environments via Gym, and considers the transfer to their visual counterparts (ALFWorld, SymTD). For reference, there is a host of perception-rich embodied environments not included in SILG due to the latter's emphasis on symbolic environments [1, 15, 32, 36, 39]. This emphasis allows SILG to provide an efficient benchmark for situated interactive language grounding. There are other complementary symbolic language grounding environments not included in this initial release of SILG due to time consideration such as [10, 45, 46]. We look forward to incorporating these in future iterations.

**Interactive language grounding** Language grounded policy-learning has been explored in the context of instruction following in tasks like navigation [8, 14, 20, 26, 31, 39, 55], games [2, 4, 21, 34, 43], and robotic control [5, 25, 52]. Touchdown, NetHack, and ALFWorld are three examples of such work included in SILG. While the above environments typically assume a small fixed set of world dynamics, other work explores settings where an agent must read text manuals to formulate appropriate policies for the game at hand. Branavan et al. [6] developed an agent to play Civilization more effectively by reading the game manual. Narasimhan et al. [40] and Zhong et al. [58] used text descriptions of game dynamics to learn policies that generalize to new environments and dynamics, without requiring feature engineering. Unlike these two works, Hanjie et al. [22] does not assume initial lexical overlap between entities in the world and entity references in the text manual. RTFM and Messenger are two examples of such work included in SILG.

**Generalization to new environments in interactive language grounding** In previous instruction following work, evaluation environments typically differ from training in their world observations. These difference range from differences in object placement in the same/new rooms (e.g. ALFWorld) to procedural generation of large game levels (e.g. NetHack). Moreover, some study generalization to new compositional instructions (e.g. Touchdown). Recent works explore generalization to new environment dynamics, which must be inferred by reading. These range from multi-step reasoning across texts (e.g. RTFM) to grounding entities to new text references (e.g. Messenger). The environments in SILG explore a variety of these generalization challenges. Many modelling techniques have been proposed to address these generalization challenges, including environmental variations [27], memory structures [29], pretrained language models [28], incremental guidance [11], subgoal-specification [3], and hierarchical RL [41]. Our baseline and analyses explores some of these techniques, including bidirectional feature-wise linear modulation [58], recurrent state-tracking [34], entity-centric convolution [34], entity-centric attention [22], and pretrained language modelling [28].

## 6 Conclusion

We introduced SILG, a new benchmark for evaluating language grounded agents across unique challenges posed by five symbolic interactive environments. Using SILG, we proposed the first shared architecture and analyzed recent methodological advancements in grounded language learning across on these environments. We showed that a shared architecture achieves comparable result to environment-specific methods, and that most advances do not result in significant gains on environments other than the one they were designed for. This highlights the need for modelling techniques that generalize across environments. Finally, the most models significantly trail human performance on SILG, which suggests ample room for future work. We hope that SILG will provide a unified platform for evaluating future methodological advances.

## Acknowledgements

We are grateful to members of UW NLP, Princeton NLP, and Facebook AI Research for their feedback, as well as the anonymous reviewers for their helpful comments and suggestions. In particular, we thank Howard Chen for detailed discussion on Touchdown and Shunyu Yao on the manuscript. Moreover, we thank Yoav Artzi, Jesse Thomason, Edward Grefenstette and Tim Rocktäschel for their invaluable feedback during the initial stages of this project. Victor is supported in part by the ARO (AROW911NF-16-1-0121) and by the Apple AI/ML fellowship. Austin is supported by the Princeton University Graduate Fellowship.

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
