## A   Impact statement

SILG facilitates research in reinforcement learning for interactive language grounding. Real-world applications in this research area range from human-computer interfaces, where users controls a computer interface via natural language specifications, to robotics control, where a robot carries out instructions given by users. Some positive impact research in this area has to do with accessibility. For example, such interfaces can allow non-experts or people to use complex software and allow people who are physically unable to operate heavy machinery to do so.

Some potential negative impact this research may have is the lack of interpretability that results from complex policies. This work uses RL, a general solution that can learn from environmental rewards without annotated data. This type of learning may result in unintuitive policies that achieve the object in surprising ways (e.g. a robot that knocks a bowl off the counter while bringing the user a cup of coffee). Language grounded policy learning, which SILG facilitates, is one way of dictating the direction of policy learning. However, more research is needed to develop more interpretable and controllable RL techniques.

The language of the individual environments in SILG are highly specific to a particular setting. For example, NetHack and RTFM are based in the fantasy settings, ALFWorld is in a household setting and Touchdown is in street navigation. This means that the learned grounding on these environments may not generalize to other settings. While SILG is an initial step, additional work is required to train general purpose agents that can interpret natural language in any setting.

## B   Using SILG

We use OpenAI Gym to create a common interface for all five SILG environments. For RTFM, Messenger, and ALFWorld, we create wrappers for the original environments such that the output of the Gym environment adheres to the shared interface. The custom environment variants we create for NetHack and for Touchdown are respective described in detail in Appendix D and E.

To instantiate a SILG environment, the user needs to simply instantiate its Gym instance as follows.

```
from silg import envs
import gym
import random
env = gym.make('silg:td_segs_train-v0', time_penalty=-0.02)
obs = env.reset()
action = random.choice(list(range(len(env.action_space))))  # e.g. 0
obs, reward, done, info = env.step(action)
```

The obs dictionary then contains the environment outputs specified in Section 2.

## C   Bidirectional Feature Wise Linear Modulation layer

Feature-wise linear modulation (FiLM), which modulates visual inputs using representations of text inputs, is an effective method for image captioning [42] and instruction following [4]. Zhong et al. [58] extends FiLM to its bidirectional variant $\text{FiLM}^2$, which they show to be effective for modeling joint multi-hop references between visual and multiple text inputs. We find $\text{FiLM}^2$ to be an effective building block for the tasks considered in SILG.

Let + and * symbols denote element-wise addition and multiplication operations that broadcast over spatial dimensions. Let $x_{\text{text}}$ denote a fixed-length $d\text{text}$-dimensional representation of the text and $X_{\text{vis}}$ the representation of visual inputs with height $H$, width $W$, and $d_{\text{vis}}$ channels. Let $\text{Conv}$ denote a convolution layer. $\text{FiLM}^2$ first modulates visual features using text features:

$$
\begin{aligned}
\gamma_{\text{text}} &= W_\gamma x_{\text{text}} + b_\gamma & (8) \\
\beta_{\text{text}} &= W_\beta x_{\text{text}} + b_\beta & (9) \\
V_{\text{vis}} &= \text{ReLU}((1 + \gamma_{\text{text}}) * \text{Conv}_{\text{vis}}(X_{\text{vis}}) + \beta_{\text{text}}) & (10)
\end{aligned}
$$

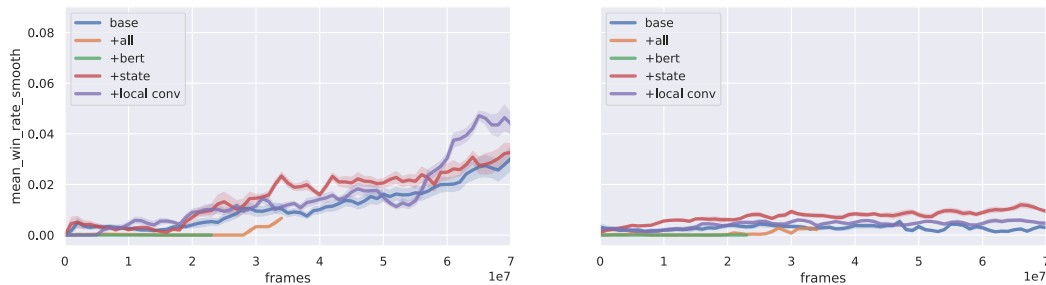

Figure 9: Manual SymTD performance. Left: train envs, right: validation envs.

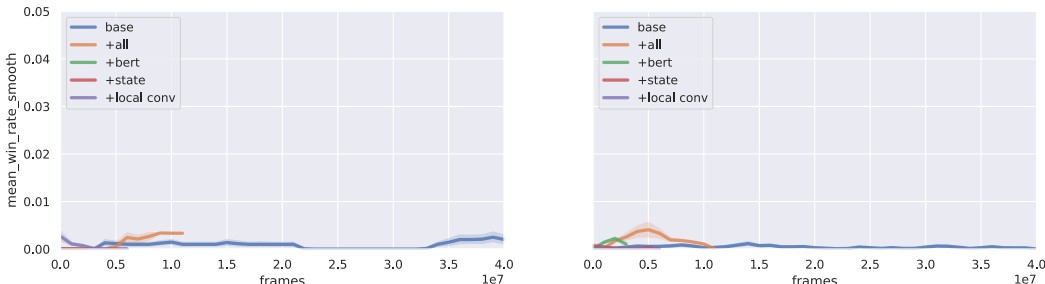

Figure 10: Manual VisTD performance. Left: train envs, right: validation envs.

Then, it modulates text features using visual features:

$$\Gamma_{\text{vis}} = \text{Conv}_{\gamma}(X_{\text{vis}}) \tag{11}$$

$$B_{\text{vis}} = \text{Conv}_{\beta}(X_{\text{vis}}) \tag{12}$$

$$V_{\text{text}} = \text{ReLU}((1 + \Gamma_{\text{vis}}) * (W_{\text{text}}x_{\text{text}} + b_{\text{text}}) + B_{\text{vis}}) \tag{13}$$

The output of $\text{FiLM}^2$ is the sum of the modulated features $V$ and its max-pooled summary $s$ across spatial dimensions.

$$V = V_{\text{vis}} + V_{\text{text}} \tag{14}$$

$$s = \text{MaxPool}(V) \tag{15}$$

## D  Multitask NetHack

For NetHack, we create a multi-task environment that uniformly samples between the three tasks `Score`, `Gold`, and `Scout`. Given the sampled task, the agent observes a text string that specifies the goal (e.g. "get more gold"), in addition to the original environment text feedback to the agent's actions. For each task, we collect 10 human playthroughs where in a human plays the original NetHack Learning Environment and attempts to get the highest score possible within 50 steps. The empirical mean of these playthroughs is then used as the task's score threshold. In the SILG version of multi-task NetHack, the agent receives a reward of 1 if it exceeds the score threshold of the current task, and 0 otherwise. If the episode terminates without exceeding the score, then the agent receives -1. We find that this method of reward assignment strikes a balance between the very different reward distributions of the individual tasks (using the raw reward from individual tasks causes the agent to only learn to play `Scout`, the dominant task with frequent rewards). NetHack does not naturally provide train/validation/test splits. We create our own splits by splitting the seed ranges (1-1,000,000 for train, 1,000,001-2,000,000 for validation, 2,000,001-3,000,000 for test).

## E  SymTD, VisTD, and Touchdown

**Navigation**    In the original Touchdown implementation, the agent navigates with left, right, and forward commands. The left and right commands rotate the panorama at the current node so that the

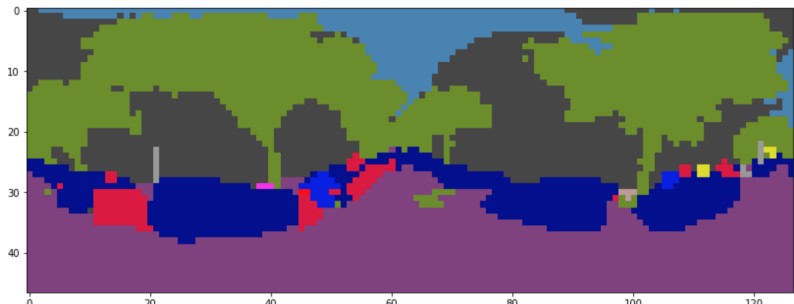

Figure 11: Segmentation map examples to create SymTD. The common objects the colours correspond to are sky (blue), buildings (gray), trees (green), sidewalk (pink), road (purple), cars (blue), traffic lights (yellow), and people (red). Note due to license agreements, this figure is a segmentation done on an example panorama provided directly on the StreetLearn website `https://sites.google.com/view/streetlearn/dataset`. Actual segmentations are visually very similar.

center of the panorama faces an adjacent node. The forward command then advances the agent to the node currently faced by the agent. We modify this navigation interface by fixing the panorama and providing the agent with a list of coordinates along the width-dimension of the panorama that corresponds to the locations of adjacent nodes that the agent may advance towards. The agent navigates by selecting one of the possible coordinates at each step. Our implementation allows the agent to see all possible navigation options upfront and reduces trajectory length by eliminating rotations. The setup is similar to [20] except our positional encoding embeds the distance of each point to the agent's current heading along the x-dimension, instead of using angle encodings.

**Rewards** Due to the sparsity of terminal $\pm 1$ rewards, we provide a reward at each step by taking the difference in shortest path graph distance before and after the step (scaled by constant factor). This does not always assign positive reward when following the gold trajectory, but we find that it is a good heuristic in most cases.

**SymTD** We pass the raw panoramas from the original Touchdown task through a PSPNet [57] trained on the Cityscapes dataset [13]. The result is a segmentation map of the raw panorama with identical height and width dimensions. To allow for caching of the segmentation maps, we downsample the segmentation maps by taking a majority vote in each $23 \times 23$ patch. We found that the majority vote caused high-frequency classes (e.g. sky) to drown out low-frequency classes (e.g. pole). Therefore, we scale the vote of each class by its inverse count computed across all segmented panoramas. If $f(c)$ is the total count for class $c$, and $P$ is a $23 \times 23$ patch in the segmentation map, the vote for class $c$ in patch $P$ is:

$$v_P(c) = \frac{1}{f(c)^\alpha} \sum_{p \in P} \mathbb{1}[p = c]$$

The representative class for each patch $P$ is then: $\max_{c \in C} v_P(c)$. We find that $\alpha = 1$ is effective at generating segmentation maps that preserve low-frequency classes. Figure 11 shows examples of such segmentation maps.

We conduct qualitative inspections of a sample of the segmented panoramas and observe that most segmentations are mostly correct relative to the input image. Despite this, human performance remains fairly low at approximately 60%. The main challenges faced by human players are (1) the symbolic features have no color information and (2) downsampling the segmentations result in highly pixelated figures, such that it is harder to distinguish smaller pedestrians from poles for example and (3) the navigation setup where the current heading is not necessarily the center of the panorama (indicated instead using an x-value) is extremely unintuitive for humans and often leads to the human player becoming disoriented. Given these observations, 60% may not be the upperbound for SymTD because controls unintuitive to humans do not affect ML models the same way.

**VisTD** We pass the raw panoramas through the ResNet-50 [23] backbone of a PSPNet trained on the Cityscapes dataset. We use the feature map from the last layer. Due to the large dimensionality

along the feature axis and the difficulty caching these for efficient RL, we reduce the number of features using PCA to the top 10 principle components. The resulting feature maps for each panorama is $47 \times 128 \times 10$.

**Manual stop TD**    In our variant of TD, the agent succeeds and the episode terminates immediately after the agent reaches the target node. We also include manual variants of SymTD and VisTD where the agent must manually select the "stop" option at the correct node. Thus, SymTD and VisTD are functionally equivalent to the original Touchdown environment.

The performance of our baseline as well as the baseline with various modelling advances are shown respectively in Figures 9 and 10 for Manual SymTD and Manual VisTD. Compared to SymTD and VisTD, the models largely fail to learn any reasonable policy within the allotted time. It remains an open question whether the complex decision process associated with manual stopping Touchdown navigation is tractable using RL, without any supervised trajectories.

## F    Collection of Human Expert Trajectories

The collection of human expert trajectories for purposes of establishing a performance upper bound is not very time-intensive. A player (paid 20$ per hour) who is familiar with text adventure games played through all five environments to collect the trajectories. Depending on the environment, the expert spent up to 30 minutes familiarizing themselves with the environment, then played approximately 50 episodes per environment, which are recorded to established human expert performance. During human playthroughs, the player is subject to the same step count limit as the RL agent. The maximum step count limit is 64 steps (for Touchdown), hence each episode is relatively quick in terms of play time.

For RTFM, Messenger, and NetHack, the human player observes a symbolic rendering of the grid along with a key that describes which symbol means which entity. The text is rendered below the grid. The human player then types in the command they would like to execute. For ALFWorld, the player observes the text rendering of the scene, as well as a list of text commands to choose from. The player then types in the index of the command they would like to execute. For Touchdown, the player observes a colour-coded rendering of the segmentation mask (of the panorama the player is in). x-coordinates are provided along the bottom of the segmentations, and a list of x-coordinates that the agent may advance towards at the next step is also provided. The player than chooses the index of the direction they would like to proceed in.

Playthrough interfaces for RTFM, Messenger, NetHack, ALFWorld, and SymTD are shown in Figure 12 through 16. Figure 11 shows examples of segmentation maps that the human player sees playing SymTD. Unfortunately we cannot include a figure of VisTD due to licensing agreement.

## G    Licenses

We distribute SILG under a MIT LICENSE, which means that researchers are free to modify and distribute our software. The environments included in SILG use their own corresponding licenses. These are

1. RTFM: Attribution-NonCommercial 4.0 International

2. Messenger: MIT

3. NetHack: NetHack General Public License

4. ALFWorld: MIT

5. Touchdown: Creative Commons Attribution 4.0 International

Of particular interest is Touchdown, whose raw panoramas come from Google Streetview. Neither we nor the creators of Touchdown distribute the panoramas. Users should follow instructions at `https://sites.google.com/view/streetlearn/dataset` to obtain the raw panoramas from Google.

```
wall            wall            wall            wall            wall            wall

wall            _               _               shimmering spear   _            wall

wall            you             _               _               _               wall

wall            _               lightning wolf  fire panther     _               wall

wall            _               _               gleaming morningstar_           wall

wall            wall            wall            wall            wall            wall

JOINT TEXT
grandmasters beat cold . gleaming beat fire . shimmering beat lightning . blessed beat poison . jaguar are order of the
 forest . panther are rebel enclave . wolf are star alliance .
FIELD TEXT
task: defeat the rebel enclave
inv:

Reward: 0        Cumulative reward: 0     Steps: 0        Done: False      Your historical scores:
Type to choose action. Type ? to see action list.
```

Figure 12: Play interface for RTFM.

```
_       _       _       _       _       _       _       _       _       _
_       _       _       _       _       _       _       _       _       _
_       _       _       _       _       _       _       _       _       _
_       _       _       _       _       _       _       _       _       _
_       _       _       airplane _      no_message _     scientist _     _
_       _       _       _       _       _       _       _       _       _
_       _       _       _       _       robot   _       _       _       _
_       _       _       _       _       _       _       _       _       _
_       _       _       _       _       _       _       _       _       _

DYNAMICS TEXT
look to the researcher, this is crucial and your main task. that motionless robot is a dangerous foe. the plane that is not moving is the restr
icted message.
NON DYNAMICS TEXT
m1: that motionless robot is a dangerous foe.
m2: look to the researcher, this is crucial and your main task.
m3: the plane that is not moving is the restricted message.

Reward: 0        Cumulative reward: 0     Steps: 0        Done: False      Your historical scores:
Type to choose action. Type ? to see action list.
```

Figure 13: Play interface for Messenger.

Figure 14: Play interface for NetHack.

Figure 15: Play interface for ALFWorld.

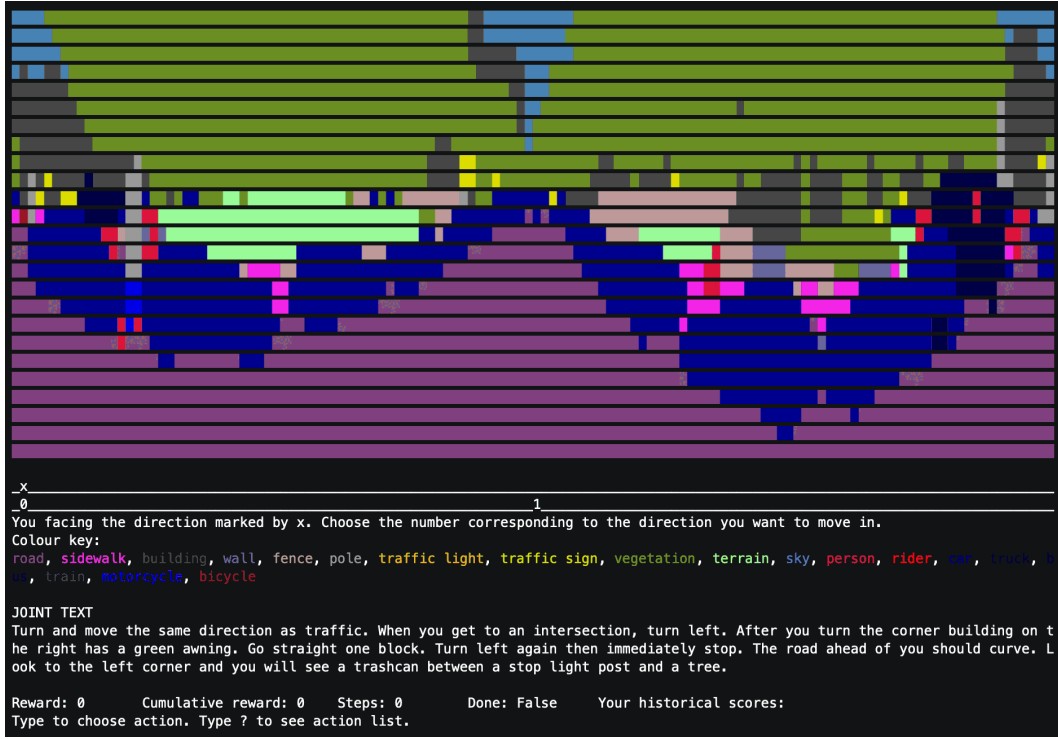

Figure 16: Play interface for VisTD.

## H  Hyperparameters

By default, we use embedding size $d = 100$, and RNN size $r = 200$. The final representation $H$ has size 400. We use 5 FiLM$^2$ layers. We train using Torchbeast [33] with an entropy cost of 0.05, baseline cost of 0.5, discount factor of 0.99, step penalty of -0.02, unroll length 80, and learning rate of 0.0005. We optimize using RMSProp with an epsilon of 0.01 and alpha 0.99. For Torchbeast parallelization, we use 30 actors, learner batch size of 24, and 4 learner threads. To account for long text sequences, we use the Huggingface PruneBERT model fine-tuned and distilled on SQuAD [47].

Due to GPU memory constraints, we reduce the model size for some environments. For NetHack, we use 30 embedding size, 100 RNN size, 8 actors, 8 batch size, and 64 unroll length. For ALFWorld, we use 10 batch size. For the Touchdown variants, we use 30 embedding size, 100 RNN size, 200 final representation size, 8 actors, 3 batch size, 64 unroll length, and 3 FiLM$^2$ layers.

## I  Compute resources

To produce our experiments, we ran 7 models each for RTFM, Messenger, and NetHack. Moreover, we ran 5 models for ALFWorld, SymTD, VisTD, Manual SymTD, and Manual VisTD. In total, this resulted in $7 \times 3 + 5 \times 5 = 46$ models. We used 5 seeds for each model, resulting in 230 runs. Each run required up to 20 CPUs (Intel Xeon) and 1 GPU (NVIDIA Quadro Pascal) for up to 2 weeks on an internal cluster. In total, we used approximately $20 \times 24 \times 2 \times 230 = 220,800$ CPU hours and $24 \times 2 \times 230 = 11040$ GPU hours.