# OpenReview forum: "SILG: The Multi-domain Symbolic Interactive Language Grounding Benchmark"
_NeurIPS.cc/2021/Conference — NeurIPS 2021 Poster_

### Official Review · Reviewer_hUgP · 2021-07-16

**Rating:** 7
**Confidence:** 4

**Summary:**

This paper presents SILG, a broad environment for language-conditioned multitask RL that unites several recently proposed symbolic language+RL environments, including RTFM, NetHack, Messenger, as well as instruction-following datasets including ALFWorld and a symbolic version of Touchdown. To unify such disparate environments, authors introduce a general framework for language conditioned RL, where agents operate given a partial observation of an environment, as well as language divided between "dynamics text", which consists of static, high-level language descriptions of the environment, and "non-dynamics text" which contains language specific to the agents' current observation and state, e.g. descriptions of the current observation, environment feedback, etc.

Authors then present a baseline model, SIR, a fairly standard language+RL model that conditions on text of varying levels of abstraction, applying successive FiLM2 layers to simulate multimodal reasoning between textual information and current observation. This entire architecture can then be trained without changes to each environment, which allows us to evaluate the generalizability of various modeling decisions in the language conditioned RL space. The authors show that many innovations proposed for certain environments (e.g. egocentric local convolution, forcing the building of representations from text dynamics) do not generalize across environments. SIR lags behind human performance on most of the environments, demonstrating potential for future work.

**Limitations And Societal Impact:**

Yes, Appendix A.

**Main Review:**

Overall, I like this paper: there is undoubtedly substantial engineering work that is put into making such disparate environments accessible under a common modeling/dataset/environment umbrella, which should certainly speed up prototyping and evaluation in language conditioned RL. I do see a few weaknesses of the paper as presented currently, however.

## Strengths

- This paper provides a broad framework for multi-task RL that conditions on language at varying levels of abstraction (dynamics/non-dynamics text). Providing a unified, simple environment for evaluation across all of these environments will undoubtedly be very useful for future work in this field.
- The paper shows quite nicely that several advancements developed for specific environments (perhaps unsurprisingly) fail to generalize reliably to other environments. This further highlights the need to come up with more general purpose language-conditioned RL techniques, in the same way that e.g. GLUE served as a good goal for more general-purpose NLU systems). Due to the unified environment and architecture, SILG should serve a similar role.

## Weaknesses

My main concerns have to do with the SymTD environment, the sparse description of the included environments, the lack of ablation studies on the base SIR model itself.

- Perhaps the most questionable part of the work is the SymTD environment, which appears to be a rather ad-hoc modification of an existing environment that has not yet been subject to the level of scrutiny of the existing environments, all of which are parts of peer-reviewed publications. There are several worrying aspects of SymTD:
    - An expert win rate of 60% is quite low and demonstrates significant noise in the environment. It could also be that many tasks have been converted incorrectly and are therefore impossible. What kinds of obstacles do human players face that result in such a low win rate?
    - Images are segmented to panoramas, but no evaluation about how accurate these panoramas are is presented. There are reasons to be suspect about the segmentation quality (e.g. because cityscapes are standard images, while touchdown is highly distorted panoramas).
- FiLM2 itself is a "recent grounded language RL modelling contribution" along the lines of those presented in 4.1, and several of the same questions asked in that section apply to the base SIR model: does FiLM2 generalize across environments? It would be good to have ablations of FiLM2, and more general ablations of the base SIR model (e.g. no language, of either kind).
- The descriptions of the environments included in SILG are extremely sparse (admittedly due to space constraints). The paper would benefit from clear examples of tasks across each environment, and a better characterization of the kinds of language information given in the environment and how they map onto the dynamics/non-dynamics text components of SIR. For example, for RTFM, presumably the game manual is categorized as dynamics text, but it's unclear whether the goal is dynamics text or non-dynamics text. Similarly, environments like ALFWorld and SymTD do *not* have high level game dynamics descriptions (I believe?)—is there simply no dynamics text in this setting? Touchdown only has one text field (L239) - is this dynamics or non-dynamics? One presentational suggestion is to add additional information in Table 1 that divides mean text len/vocab size by whether or not it's treated as dynamics text or not, and additionally a high level description of what kinds of language belong in which category (e.g. RTFM has game manual, ALFWorld has text description of environment, etc). (Maybe in Figure 1 as well?)
- Along these lines, the "relative position" feature for SIR is only briefly mentioned in L141, which left me confused. Why is this relative position to "a particular cell"? What particular cell is this, and why is it necessary?
- Some environments may be too easy: the inclusion of only the first stage of RTFM is rather odd, since we are already at human performance on this baseline (Table 2). What was the motivation for including only the first stage? (Similarly, only an easier subset of tasks was selected for Messenger and NetHack). Are the more difficult stages in these papers still compatible with both the SILG environment and the SIR framework? I would hope that authors will be able to include such harder environments in subsequent releases.

## Questions

- This was actually not super clear from the paper: SIR is a unified architecture, but it is trained in isolation on each individual environment, correct? Rather than training a single model across all environments.
    - (Assuming the above is true) SILG might enable some interesting questions regarding training across multiple environments and transfer learning. E.g. perhaps training on some environments actually enables faster learning/transfer to others, since there are general game-playing ideas common to each environment. Did authors try the (dramatic) attempt of training jointly across all environments at once?
- Table 2 lists performance of the best variant for each environment—what is the variant that leads to the best performance averaged across all environments? Presumably it's "+state"?

**Time Spent Reviewing:**

3

---

> ### Author Response · Authors · 2021-08-07
> **Author response**
>
> We thank the reviewer for their detailed comments. We are glad that you found our benchmark and ablations useful. We are also excited about the potential for SILG as a language-conditioned RL testbed for unified architectures. We especially appreciate the discussion regarding SymTD as well as joint training and cross-environment transfer. Please find our detailed response below.
>
> *1. (RE SymTD) An expert win rate of 60% is quite low and demonstrates significant noise in the environment. It could also be that many tasks have been converted incorrectly and are therefore impossible. What kinds of obstacles do human players face that result in such a low win rate?*
>
> The main challenges faced by human players are (1) the symbolic features have no color information and (2) downsampling the segmentations result in highly pixelated figures, such that it is harder to distinguish smaller pedestrians from poles for example and (3) the navigation setup where the current heading is not necessarily the center of the panorama (indicated instead using an x-value) is extremely unintuitive for humans and often leads to the human player becoming disoriented. Given these observations, 60% may not be the upperbound for SymTD because controls unintuitive to humans do not affect ML models the same way. However, we do expect that in the limit of more compute/time and better models, VisTD performance should exceed SymTD due to more complete perception. Our current results suggest that SymTD is a more efficient environment to learn in. One interesting follow-up question is whether SymTD can be used to bootstrap VisTD, resulting in even higher performance while maintaining learning efficiency.
>
> *2. (RE SymTD) Images are segmented to panoramas, but no evaluation about how accurate these panoramas are is presented. There are reasons to be suspect about the segmentation quality (e.g. because cityscapes are standard images, while touchdown is highly distorted panoramas).*
>
> The quality of the segmentations are quite reasonable (unfortunately we do not have human labels to cross-check accuracy, but will try to quantify this in the future), and are good from qualitative inspection despite the panoramas being slightly distorted at the edges. We will release segmentation results alongside SILG, Touchdown copyright permitting.
>
> *3. FiLM2 itself is a "recent grounded language RL modelling contribution" along the lines of those presented in 4.1, and several of the same questions asked in that section apply to the base SIR model: does FiLM2 generalize across environments? It would be good to have ablations of FiLM2, and more general ablations of the base SIR model (e.g. no language, of either kind).*
>
> Thank you for this suggestion. We will add these ablations to the manuscript.
>
> *4. The descriptions of the environments included in SILG are extremely sparse (admittedly due to space constraints). The paper would benefit from clear examples of tasks across each environment, and a better characterization of the kinds of language information given in the environment and how they map onto the dynamics/non-dynamics text components of SIR... One presentational suggestion is to add additional information in Table 1 that divides mean text len/vocab size by whether or not it's treated as dynamics text or not, and additionally a high level description of what kinds of language belong in which category (e.g. RTFM has game manual, ALFWorld has text description of environment, etc). (Maybe in Figure 1 as well?)*
>
> We will add a table with detailed description and examples that outline what dynamics and non-dynamics texts are for different environments. For RTFM and Messenger, the manual is the dynamics text and everything else is non-dynamics text. For ALFWorld, we concatenate the text fields (initial goal, current observation for ALFWorld) and treat this concatenated text as dynamics text. The individual components are used as non-dynamics text for the purpose of attending to the concatenated text. For NetHack and SymTD, which only have one text field, this field is considered to be both dynamics and non-dynamics text and the model essentially computes self-attention.
>
> *5. Along these lines, the "relative position" feature for SIR is only briefly mentioned in L141, which left me confused. Why is this relative position to "a particular cell"? What particular cell is this, and why is it necessary?*
>
> Relative position is a cell-wise feature that denotes the position of the cell relative to the player agent. This feature informs the model how far and in which direction an observation is relative to the agent’s position.
>
> *6. Some environments may be too easy: the inclusion of only the first stage of RTFM is rather odd, since we are already at human performance on this baseline (Table 2). What was the motivation for including only the first stage? (Similarly, only an easier subset of tasks was selected for Messenger and NetHack). Are the more difficult stages in these papers still compatible with both the SILG environment and the SIR framework? I would hope that authors will be able to include such harder environments in subsequent releases.*
>
> The difficult stages were not included in this manuscript because making progress on them require curriculum learning (at least according to current literature), and its presentation would greatly complicate the description of SILG and SIR. We agree that including these more difficult stages is useful to the community. We will include subsequent difficult stages of RTFM and Messenger, as well as manual stop Sym/Vis TD in our release.
>
> *7. This was actually not super clear from the paper: SIR is a unified architecture, but it is trained in isolation on each individual environment, correct? Rather than training a single model across all environments.*
>
> That is correct. We will clarify this in the manuscript.
>
> *8. (Assuming the above is true) SILG might enable some interesting questions regarding training across multiple environments and transfer learning. E.g. perhaps training on some environments actually enables faster learning/transfer to others, since there are general game-playing ideas common to each environment. Did authors try the (dramatic) attempt of training jointly across all environments at once?*
>
> We agree that cross-environment transfer is a worthwhile investigation which we did not include due to resource constraints. One intuition is that while the environments require different action spaces, certain concepts such as spatial movements, relative position references, enemies are shared between environments. Hence, perhaps performance in one environment is transferable to another. We would like to investigate this in future work.
>
> *9. Table 2 lists performance of the best variant for each environment—what is the variant that leads to the best performance averaged across all environments? Presumably it's "+state"?*
>
> That is correct. We will add a statement to the manuscript to reflect this.

---

> > ### Comment · Reviewer_hUgP · 2021-08-10
> > **Response**
> >
> > Thanks to the response to my review! I have some hesitations about SymTD, and would be very interested in the FiLM2 ablation studies, but don't think these are crucial weaknesses. I've read the other reviews and responses, and remain satisfied with my score of 7.

---

### Official Review · Reviewer_tPsp · 2021-07-16

**Rating:** 7
**Confidence:** 4

**Summary:**


This paper is about a novel benchmark to evaluate interactive language-grounding models called Symbolic Interactive Language Grounding benchmark (SILG). It regroups five different grounded language learning environments, each with its own key challenges. To perform well on SILG, a model would need to generalize to new dynamics, entities, and partially observed worlds, understand rich natural language and perform multi-step reasoning. The authors designed a common interface for researchers to evaluate their models across all of these environments. In the paper, in addition to describing SILG, the authors propose the first architecture, Symbolic Interactive Reader (SIR), that can be used for all of these environments. Experiments were conducted on SILG to study how several recent advances in language-conditioned RL generalize over these environments (FiLM, recurrent state-tracking, large pretrained LMs, etc). Empirically, it was shown that most of the proposed components are only useful for the environment they were proposed for.

**Limitations And Societal Impact:**

The authors did discuss how they have adapted each environment to fit under a common API (e.g., convert text description to world representation). They also mentioned some issues human experts had with SymTD due to the symbolic representations discarding information such as color.

The authors have included a thorough impact statement in the appendix. They discussed the lack of interpretability that results from complex policies, especially those learned via RL (learning may result in unintuitive policies that achieve the object in surprising ways).

**Main Review:**


## Overview

The proposed benchmark, SIGL, is a valuable contribution to the research community to track progress on interactive language-grounding. To the best of my knowledge, the proposed model, SIR, is novel and offers a good first baseline to compare against on this benchmark. The authors clearly described what are the key challenges and properties of each environment in SIGL. To me, this submission appears to be technically sound and each engineering decision is well motivated. The paper is easy to follow and well-written. I believe the paper provides enough information for me to build the model (Figure 2, Section 3, and Supplementary material) and try it on SIGL (once release). Overall, I agree there's a need for unifying environments used by the interactive language-grounding research community. Having standardized benchmarks help tracking research progress. I recommend this paper for acceptance.


## What I like about this paper

- The proposed benchmark unifies five language-grounding environments under a common interface. SIGL allows to quickly test new methods and see how much progress the community is making.
- The ablation study on the different recent advances in language-conditioned RL and how useful they are for environments other than the one they were proposed for.
- Table 1 about SIGL statistics showing the distinctive key properties for each environment.


## Concerns

- p.6 line 198: State-tracking does not help generalize to the evaluation distribution.

- Fig.6 seems to be missing +bert and +all lines.

- In ALFWORLD, "the names of which we concatenate into a symbolic world observation grid". I don't understand how the symbolic world observation grid looks like exactly.

-----
## Typos
- p.4: line 109: "is not complex, but [observation?] are 100 words on average", missing word?


**Time Spent Reviewing:**

3

---

> ### Author Response · Authors · 2021-08-07
> **Author response**
>
> We thank the reviewer for their detailed comments. We are glad that you found our benchmark and ablations useful. Please find below our responses.
>
> *1. p.6 line 198: State-tracking does not help generalize to the evaluation distribution.*
>
> State tracking helps fit to train faster, but because of the adversarial nature of the train-eval split on Messenger, this doesn't help with generalization on eval.
>
>
> *2. Fig.6 seems to be missing +bert and +all lines.*
>
> These experiments were not finished at the time of submission (but the partial results (which unfortunately are partially obstructed in Fig 6) underperform similar to Figure 7). We will add the now completed result (which still shows +bert and +all underperforming) to the manuscript.
>
> *3. In ALFWORLD, "the names of which we concatenate into a symbolic world observation grid". I don't understand how the symbolic world observation grid looks like exactly.*
>
> In ALFRED, a scene is composed of several objects (e.g. cup 1, bottle 2). We build an artificial grid using this list of objects where each entry of the grid contains the string for an object. For example, given these two objects, the grid would consist of one row. The first entry of the row would be “cup 1”. The second entry would be “bottle 2”. The rest of the model would work as usual. We will clarify this in the manuscript.

---

> > ### Comment · Reviewer_tPsp · 2021-09-14
> > **Thank you for the response**
> >
> > It seems my reply acknowledging your response didn't post properly!
> >
> > From memory, what I wanted to say:
> >
> > I remain satisfied with my review. I believe this framework is going to be useful and look forward to seeing improvements/addition in the future.

---

### Official Review · Reviewer_zwvr · 2021-07-16

**Rating:** 7
**Confidence:** 4

**Summary:**

The paper provides a benchmark that wraps multiple interactive language grounding environments in a shared symbolic environment paving the way for agents that can benchmark on multiple different tasks at once. A further shared architecture baseline, the Symbolic Interactive Reader, is benchmarked on each of the individual environments and is further ablated with respect to its components (state tracking, entity centric convolutions+attention, and pretrained language models).

**Ethical Concerns:**

There are no immediately applicable ethical concerns with this work.

**Limitations And Societal Impact:**

There is no Broader Impacts section provided. A brief section on the biases of the language present in the individual environments could be added given that the environments are trying to learn grounded language from these environments with potential for other downstream tasks.

**Main Review:**

Pros:

The SILG benchmark itself is easily usable in the form of Gym wrappers.

SIR provides a unique baseline and an initial method for other researchers to build off of in creating such combined models.

The paper was clearly written and the choices for the architecture and all of the individual environments were justified.


Cons/Questions:

Its not immediately clear to me why we want an architecture for NetHack to do well in ALFWorld. They are both trying to ground language in environment mechanics, yes, but the core issues faced are different. ALFWorld has much more of a focus on language (richer descriptions, text -> vision transfer, etc.) while NetHack is all about generalizing to unseen environment structures (a natural consequence of the procedural generation, without nearly as rich a language component). RTFM <-> NetHack or ALFWorld <-> Touchdown makes more immediate sense. A discussion regarding exactly what properties these individual environments are trying to foster in RL agents would appear (and are desirable) in a SILG environment agent would go a long way towards tying this paper together (expand on the Table 1 key challenges line perhaps).

As an RL benchmark, it would be really useful to have environment only benchmarks such as FPS and if any FPS is lost/gained over the underlying environments.

**Time Spent Reviewing:**

3

---

> ### Author Response · Authors · 2021-08-07
> **Author response**
>
> Thank you for the detailed comments. We are glad you find our benchmark and baseline useful, and the manuscript clearly written and decisions justified. We will add more details to the manuscript regarding the challenges of each environment, as well as the efficiency of each environment (e.g. FPS). Please find our detailed responses below.
>
> *1. Why we want an architecture for NetHack to do well in ALFWorld. They are both trying to ground language in environment mechanics, yes, but the core issues faced are different. ALFWorld has much more of a focus on language (richer descriptions, text -> vision transfer, etc.) while NetHack is all about generalizing to unseen environment structures (a natural consequence of the procedural generation, without nearly as rich a language component). RTFM <-> NetHack or ALFWorld <-> Touchdown makes more immediate sense.*
>
> We agree that the core issues faced in Nethack and ALFWorld are different. Ideally, we would like an architecture that is able to perform well given the issues faced in both settings (e.g. richer descriptions and diverse partially observed environment structures). For example, an ideal proposed method should improve performance in both settings; a slightly less ideal method should improve performance in one without hurting the other.
>
> *2. A discussion regarding exactly what properties these individual environments are trying to foster in RL agents would appear (and are desirable) in a SILG environment agent would go a long way towards tying this paper together (expand on the Table 1 key challenges line perhaps).*
>
> We have some details regarding the challenges in each environment in Section 2. We will add more details to the manuscript by expanding Table 1.
>
> *3. As an RL benchmark, it would be really useful to have environment only benchmarks such as FPS and if any FPS is lost/gained over the underlying environments.*
> We will add this description to the manuscript for each environment. The FPS are RTFM 240, Messenger 1627, Nethack 439, SymTD 779, ALFWorld 7. These numbers are obtained using a random agent without parallelization (e.g. without Torchbeast) on a machine with 500 GB RAM and Intel(R) Xeon(R) CPU E5-2698.
>
> *4. There is no Broader Impacts section provided. A brief section on the biases of the language present in the individual environments could be added given that the environments are trying to learn grounded language from these environments with potential for other downstream tasks.*
>
> We will add a broader impact statement to the manuscript detailing the limitations of SILG environments and how they may limit downstream applications.

---

> > ### Comment · Reviewer_zwvr · 2021-09-01
> > **Reply**
> >
> > Thanks to the authors for the response. Most of my (immediately fixable) concerns have been answered and I will keep my score.

---

### Official Review · Reviewer_bubM · 2021-07-16

**Rating:** 4
**Confidence:** 5

**Summary:**

Because existing language grounding work operates in single environments, it is difficult to determine whether model contributions apply across diverse settings. To remedy this, the authors present the Symbolic Interactive Language Grounding Benchmark (SILG), an environment simplifying and unifying five existing language grounding environments: RTFM, Messenger, NetHack, ALFWorld, and SymTD. The authors focus on symbolic environments over visual environments to abstract out the component of visual perception.
In addition, the authors provide a baseline model for SILG, Symbolic Interactive Reader (SIR). SIR's core architecture is compatible with all environments in SILG. They test SIR and several variants thereof containing recent modelling contributions, finding that while SIR achieves decent performance on the benchmark, it falls short of human expert playthroughs, suggesting room for future work. In addition, they find that most contributions except for recurrent state tracking are beneficial in only the environment for which they were originally developed.

**Ethical Concerns:**

No ethical concerns.

**Limitations And Societal Impact:**

Societal impact is well-addressed.

**Main Review:**

Contributions:
1. Introducing a multi-environment symbolic language grounding benchmark, SILG that combines five language-grounding environments under the same interface.

While the authors make a good argument for training/testing models on multiple, diverse environments instead of single environments, the five complex existing environments are significantly simplified. The value added for combining these existing benchmarks under one umbrella is unclear.


2. Providing a baseline model for this benchmark (SIR).

SIR consists of a FiLM^2 model which is provided with different types of inputs. The authors should simply say they use FiLM^2 as the baseline model.


3. Results of variants of FiLM^2 (termed SIR) on these benchmarks.

It’s unclear what insights are gained from these results which would lead to better models. This is underscored by the fact that it’s hard to compare performance across such different benchmarks on different models.


===============================================================================

High-level review

i. Testing on multiple, diverse environments to analyze generalization of RL algorithms is an existing paradigm. It is not clear why we want to combine simplified versions of five existing environments into one benchmark. It’s also detrimental to keep the original names like (NetHack) for the simplified version; this will only sow confusion.

ii. Generalization is the main focus, but different models are used for different experiments and no cross-environment generalization is presented. This weakens the results significantly.

iii. Fundamental details are missing, such as what the dynamic text is for different games, making the results impossible to reproduce.

iv. A core result of the paper suggests that "algorithm X works on environment A but not environment B". This is misleading, as models are not comparable across environments (model size, environment size, hyperparameters).


More in the detailed review.

===============================================================================

Testing the generalization of existing algorithms to other environments is an interesting problem. However, in general, the method, model selection, and evaluation need to be better described and justified. There are several major issues with the manuscript as it stands, summarized here from the detailed review below with references to the detailed review:

- (1, 2, 12) The authors should discuss how to interpret the analysis of different model enhancements. It's not clear that different models, environments, and tasks are comparable, especially when the base models have different hyperparameters for different environments, and when models are trained on single environments. This reduces the impact of combining multiple environments into one benchmark. If the result is that models are trained/tested on single environments, it is not clear that the authors have demonstrated significant progress towards solving the motivating problem.

- (3, 5, 6, 11) As it stands, the work is not reproducible from reading the paper. The authors should report methodology with more detail, including environment simplification and model selection.

- (4, 8, 10) Reported results (Tables 2, 3) are difficult to interpret and lacking some information.

I would vote for rejecting the publication based on the comments above.

========================================================================================

Detailed comments:

Major:

1) Why can we compare the algorithmic contributions across different environments? There is no relationship between the environments, and the models are different for each environment. This makes results very difficult to interpret.

2) The evaluation metric in comparing the performance of recent modelling contributions is not stated (it would help to add a couple words on the evaluation metric at the beginning of Section 4.1). I believe the evaluation metric is an algorithm's performance improvement over the base.

3) (Appendix G) How are model hyperparameters selected? Are learning hyperparameters the same for each environment? Tuning hyperparameters for each environment separately results in a different model per environment.

4) Standard deviations in Tables 2, 3 are quite large and overlapping, which makes it hard to differentiate Base and Best.

5)  Pg. 3 (NetHack): There are 23 fixed actions (Table 1). However, in the NetHack paper, there are 93 possible actions that players can take. How is the subset of actions chosen? In the expert playthroughs, do they also only access the 23 same fixed actions?

6) Pg. 3 (NetHack) "…in our expert playthroughs of SILG NetHack, we observe just over 100 unique words." How is this number relevant to the agent? How does the agent name objects?

7) Lines 30-32, Figure 1, lines 57-58 state that SILG contains five language-grounding environments (RFTM, Messenger, NetHack, ALFWorld, SymTD). However, SILG does not contain RFTM nor NetHack, but rather one level of RFTM and three NetHack tasks. Like ALFWorld and SymTD, it is important to avoid stating that SILG contains RFTM and NetHack. This could be remedied by giving the simplified RFTM and NetHack versions a different name and stating their relationship with the original RFTM, NetHack environments.

8) Please include the standard deviation for SOTA in Table 2 so that Base, Best, and SOTA are easier to compare.

9) "Table 3 shows that policies learned in the symbolic environment transfer to the 3-D environment."-- For sake of comparison, the SOTAs on ALFRED and VisTD should be included.

10) Though one of the main conclusions of the paper is that recurrent state-tracking is beneficial across environments (including SymTD), it does not result in the best transfer results to VisTD. If transfer to VisTD is the main practical usage of training with SymTD, then finding that recurrent state-tracking is best in SymTD may not be relevant.

11) A key challenge in reproducibility is that we don't know what the dynamics text and non-dynamics text are for each environment. It would be helpful to state for each environment where these texts are generated or found in instruction manuals, how they're processed, etc.

12) Models differ per environment (thus incomparable) and are trained/tested on single environments. Training/testing on single environments was a motivation for the work. Moreover, environments are simplified versions of the originals. Then, how does using SILG improve from training individually on the original environments?

Minor:

14) How was the experimental subject recruited? If possible, it would be better to have more participants.

15) "This may be because it helps prevent local minima that cause repetitive actions". I would look into pg. 2 in the Resnet paper (https://arxiv.org/pdf/1512.03385.pdf) for a theoretical explanation for why H' may converge faster than $LSTM(H, S_{t-1})$.

16) "Dynamics" and "non-dynamics text" is confusing terminology. It may help to italicize these to help the reader view them as being defined in lines 66-67.

17) Line 160: how do we get 4 in the subscript?

18) Equations 5, 6: Perhaps replace the C, A in the subscripts or remove the subscripts? As they were being defined on the left and reused in the subscript.

19) Reference hops are mentioned several times throughout the paper, and are estimated using the number of sentences and "then"s in textual inputs. How do reference hops relate to how good a model is?


Typos:

Appendix D: principle -> principal

Appendix C: dtextdimensional -> d-dimensional


**Time Spent Reviewing:**

13

---

> ### Author Response · Authors · 2021-08-07
> **Author response**
>
> We thank the reviewer for their detailed comments. We are glad that you agree with our thesis of training and testing models on multiple diverse environments. Regarding the central concern that we train separate models on each environment: while cross-environment generalization is an important goal, we also need to find modelling techniques or representations that work well for multiple environment. Aggregate benchmarks such as GLUE helped demonstrate that better representations improve many tasks without cross-task training and generalization. This was an important contribution to the community and an inspiration for this work. It may work similarly in language grounding, where representations as opposed direct cross-environment generalization is more important. SILG is well-positioned to demonstrate this. Moreover, while we include simplified TD and the first curriculum stages of RTFM and Messenger in this work, we show that they pose a challenge to existing models. We will incorporate more difficult versions of environments in our release. Regarding reproducibility: we submitted our source code for SILG and SIR and experimental setup in the supplemental materials, and will also open-source them for public access. We will also rename SILG NetHack and mark curriculum stages of RTFM and Messenger to avoid confusion with existing environments. Please find below our detailed responses. We hope that the reviewer finds our responses satisfactory, that SILG is a beneficial addition to the community, and considers increasing their score.
>
> *(Response to high level points i, ii) 1. Five complex existing environments are significantly simplified... Keeping same name for simplified envs sows confusion.*
>
> Our goal is to provide a benchmark for researchers to test a common model or algorithm for language grounding across a diverse set of tasks, which we believe SILG accomplishes. To clarify, SILG does not simplify environments, apart from arguably Touchdown. The RTFM and Messenger environments in our experiments are stage 1 of the original curriculums provided in their respective papers. In SILG, we multitask multiple NetHack tasks (unclear whether this is a simplification). The initial findings in ALFWorld was that using precomputed commands were not scalable despite having supervised trajectories. Our work makes effective use of precomputed commands despite not using supervised trajectories. For Touchdown, we make the simplification of auto-stopping, however we again do not use supervised trajectories and instead do RL using proximity to the target location as reward. As our findings show, these included environments pose a significant challenge to existing methods. We view SILG as an evolving benchmark and plan to include harder variants of RTFM, Messenger, and Touchdown in future releases. We will modify the NetHack name so as to avoid confusion.
>
> *2. Baseline is FiLM^2*
>
> The baseline is a generalization of FiLM^2 that supports arbitrary instance-specific text (non-dynamics text) and instance-agnostic rule text (dynamics text). The FiLM^2 model assumes that the environment has manuals, goal, and inventory, which is not the case in general. Moreover, our baseline uses the same form of dynamics/non-dynamics attention regardless of the non-dynamics text field, unlike FiLM^2 which uses different hand-crafted attention for goal vs. inventory. Finally, our baseline accommodates the different types of action spaces in SILG.
>
> *3. Unclear what insights are gained from results that would lead to better models - hard to compare performance across such different benchmarks on different models...*
>
> Please see 5.
>
> *4. Fundamental missing details such as what the dynamic text is for different games make results impossible to reproduce*
>
> Please see 15.
>
> *(response to ii, iv) 5. Why can we compare the algorithmic contributions across different environments? There is no relationship between the environments, and the models are different for each environment. This makes results very difficult to interpret.*
>
> Sorry that the term “generalize” is ambiguous - we will consider alternative naming. Our goal is to facilitate the investigation of architectures and methodology that improve performance on a diverse set of environments with unique challenges. We illustrate that SILG indeed allows for these types of study by investigating a particular architecture (SIR) and characterizing its performance across environments. SIR is an architecture that is shared across environments, with minimal modifications to accommodate differences in action space (similar to recent benchmarks like GLUE where the output space is different for each task). Jointly training a model on all environments is a worthwhile direction and is enabled by SILG.
>
> *6. The evaluation metric in comparing the performance of recent modelling contributions is not stated (it would help to add a couple words on the evaluation metric at the beginning of Section 4.1)...*
>
> We evaluate using win-rate at time of convergence as well as learning curves (e.g. how efficient is win-rate vs. frames). We will add the win rate improvement over the baseline to the manuscript in an additional table.
>
> *7. (Appendix G) How are model hyperparameters selected? Are learning hyperparameters the same for each environment? Tuning hyperparameters for each environment separately results in a different model per environment.*
>
> We sweep hyperparameters for each environment, which results in slightly different hyperparameters for each task (listed in appendix H). This indeed results in different models, but the core architecture is the same. All our code will be released.
>
> *8. Standard deviations in Tables 2, 3 are quite large and overlapping, which makes it hard to differentiate Base and Best.*
> There is indeed overlap in terms of the converged win-rate for a number of environments, however the learning curves (e.g. Figure 3-7) show that some variants are much more efficient than others.
>
> *9. Pg. 3 (NetHack): There are 23 fixed actions (Table 1). However, in the NetHack paper, there are 93 possible actions that players can take. How is the subset of actions chosen? In the expert playthroughs, do they also only access the 23 same fixed actions?*
>
> We don’t modify the action space from the tasks in the Nethack Learning Environment (NLE). While NLE technically supports 93 possible actions, the tasks proposed for NLE (e.g. score, stairs, pets, gold, food, scout) only have 23 actions. One can verify this by instantiating NLE’s gym interface and confirming that `env.action_space` is `Discrete(23)`. This smaller set of task actions are specified in https://github.com/facebookresearch/nle/blob/master/nle/env/tasks.py. The expert playthroughs also only access these 23 instructions
>
> *10. Pg. 3 (NetHack) "…in our expert playthroughs of SILG NetHack, we observe just over 100 unique words." How is this number relevant to the agent? How does the agent name objects?*
>
> This number is the number of unique word types the player will see when playing through NLE subtasks. While the overall Nethack vocabulary is undoubtedly larger than this, the agent typically only encounters a much smaller number of word types (e.g. in the form of in-game messages, goal statements) in its exploration
>
> *11. Lines 30-32, Figure 1, lines 57-58 state that SILG contains five language-grounding environments (RFTM, Messenger, NetHack, ALFWorld, SymTD). However, SILG does not contain RFTM nor NetHack, but rather one level of RFTM and three NetHack tasks. Like ALFWorld and SymTD, it is important to avoid stating that SILG contains RFTM and NetHack. This could be remedied by...*
>
> We will modify the manuscript to more clearly state that we only use the first part of RTFM and Messenger curriculum. We will also include difficult stages of curriculum in our release.
>
> *12. Please include the standard deviation for SOTA in Table 2 so that Base, Best, and SOTA are easier to compare.*
>
> We will add this where possible (e.g. the original work reported standard deviation). The standard deviation for RTFM is 21, Messenger is 1.4, and ALFWorld imitation learned on supervised trajectories did not measure variance.
>
> *13. "Table 3 shows that policies learned in the symbolic environment transfer to the 3-D environment."-- For sake of comparison, the SOTAs on ALFRED and VisTD should be included.*
> We will modify the manuscript to include this.
>
> *14. Though one of the main conclusions of the paper is that recurrent state-tracking is beneficial across environments (including SymTD), it does not result in the best transfer results to VisTD. If transfer to VisTD is the main practical usage of training with SymTD, then finding that recurrent state-tracking is best in SymTD may not be relevant.*
>
> Although state tracking did not improve VisTD directly, it improved SymTD and therefore VisTD indirectly. The SymTD performance is directly transferable to VisTD, because one can run segmentation on VisTD to obtain SymTD, then run the policy learned in SymTD.
>
> *(response to iii) 15. A key challenge in reproducibility is that we don't know what the dynamics text and non-dynamics text are for each environment. It would be helpful to state for each environment where these texts are generated or found in instruction manuals, how they're processed, etc.*
>
> We will add a table with detailed description and examples that outline what dynamics and non-dynamics texts are for different games. Our source code and experiment setup were submitted for review as a part of the supplementary materials and will also be open-sourced for reproducibility.
>
> *16. Models differ per environment (thus incomparable) and are trained/tested on single environments. Training/testing on single environments was a motivation for the work. Moreover, environments are simplified versions of the originals. Then, how does using SILG improve from training individually on the original environments?*
>
> Please see 5.

---

> > ### Author Response · Authors · 2021-08-08
> > **Author response to minor points**
> >
> > *How was the experimental subject recruited? If possible, it would be better to have more participants.*
> >
> > The experimental subject is a CS student trained to be proficient in these games.
> >
> > *"Dynamics" and "non-dynamics text" is confusing terminology. It may help to italicize these to help the reader view them as being defined in lines 66-67.*
> >
> > We'll make this modification to the manuscript.
> >
> > *Line 160: how do we get 4 in the subscript?*
> >
> > That's a typo that we will correct. Linear_4 is just to denote that it does not share parameters with the other linear layers.
> >
> > *Equations 5, 6: Perhaps replace the C, A in the subscripts or remove the subscripts? As they were being defined on the left and reused in the subscript.*
> >
> > We'll make this modification to the manuscript.
> >
> > *Reference hops are mentioned several times throughout the paper, and are estimated using the number of sentences and "then"s in textual inputs. How do reference hops relate to how good a model is?*
> >
> > It's not a measure of how good a model is but of how many reasoning steps is required for a human player.

---

> ### Author Response · Authors · 2021-08-24
> **Do you have any remaining questions?**
>
> Hi, does the reviewer have any remaining questions you would like us to address? Have we addressed all your previous concerns? If we have addressed the reviewer’s concerns, would you please consider increasing your score?

---

> > ### Comment · Reviewer_bubM · 2021-08-30
> > **Response**
> >
> > Thanks for your very thorough response! I'm going to increase my score to 4 after reading the other reviews and responses for the following reasons:
> >
> > My major concerns mainly fall into clarity and thesis, and with the new modifications, clarity is no longer a major concern. However, since the goal is "to facilitate the investigation of architectures and methodology that improve performance on a diverse set of environments with unique challenges", it's still not clear what we gain from simplifying 5 different environments under one framework vs using the original environments, especially when the models are trained on environments separately in this paper.

---

> > > ### Author Response · Authors · 2021-08-30
> > > **Thank you**
> > >
> > > Thank you for your response! The environments included in SILG have different observation and action spaces. As a result, prior work in this area tend to develop model against a single environment and do not evaluate against other environments. As reviewers zwvr, rPsp, and hUgP note, in SILG we put in significant engineering work to standardize these environments under an easily usable Gym interface, abstracting out perceptual differences while maintaining key language grounding challenges. **The gain from SILG is that researchers can easily test and ablate new methods (such as our experiments with SIR) against the challenges posed in each environment, without significant code modifications.** That we are the first work to train/evaluate on these environments with the same architecture also speaks to the difficulty of building a shared interface for these environments and supports the importance of unifying these environments with SILG.

---

### Decision · Program_Chairs · 2021-09-28

**Decision:**

Accept (Poster)

**Comment:**

Codifying benchmarks and bringing them under a single umbrella encourages better research. Particularly in an area like RL where results are often tenuous, depending on factors like random seeds. Larger and more comprehensive benchmarks ensure that methods really do improve on the state of the art.

While the idea is good, two major concerns came up during the discussion.

1. The work seems incomplete.

None of us could understand why the authors include only partial variants of existing benchmarks when creating SILG. This seems like a significant weakness that limits the SILG's contribution. The authors cite GLUE as their inspiration, but GLUE included other benchmarks wholesale; one didn't need to separately evaluate on them anymore because the final GLUE result already did so. By not including all of the games, as they were meant to exist, the authors sell their own approach short. We would love to see SILG published, with the full games included.

The authors put forward that they only use partial games because current methods can't deal with full games anyway and the resulting evaluation numbers would be too poor. Many methods exist for dealing with this, for example, in the retrieval community, methods usually don't retrieve the top result with perfect accuracy, and so we measure top-n (R@5, R@10, etc.) retrieval accuracy. The authors could do the same. Include the full games and then measure performance at 1% of the game, 10%, 20%, etc. This would make SILG a far more lasting contribution.

2. Essential details, that determine the technical validity of the benchmark and of the results, are completely missing.

Since the authors cite GLUE as their inspiration, it's worth having a second look at that paper. It very carefully defined what the inputs, outputs, and metrics are for every single benchmark rolled into GLUE. Unfortunately, almost all details to do with how models interact with the games are missing here.

Including such details is essential. They make the difference between a technically correct and a flawed manuscript. Imagine that the authors had crafted a mapping to one of the games in such a way that the game is now unplayable or far more difficult than it normally is. This is very easy to do.

As it stands, reviewers could not evaluate the technical correctness of the manuscript.

Even in terms of the novel conceptual contribution of the manuscript, it's really this mapping that we should be evaluating. It's the core of the method that brings the games together, and it’s missing.

One more minor point, the human experiment was generally seen as inadequate having only one subject.

I encourage the authors to continue with SILG despite this unfortunate news. The resulting benchmark and manuscript will be far more useful and impactful when fleshed out fully and when presented with enough detail to both reproduce the work and judge its technical correctness.

**Consistency Experiment:**

NeurIPS has a long history of experimentation. In 2014, NeurIPS ran an experiment in which 10% of submissions were reviewed by two independent committees to quantify the randomness in the review process. This year, we repeated a variant of this experiment to see how the quality of the review process has changed over time.  This paper was part of the experiment and was therefore assigned to two committees (consisting of reviewers, an Area Chair, and a Senior Area Chair) that reached independent decisions.  If both committees made the same recommendation, this recommendation was followed. If a single committee recommended acceptance, the paper was accepted (with the exception of a few cases in which the other committee identified what we considered a fatal flaw, e.g., an error in a key result).

This copy’s committee reached the following decision: **Reject**

The other committee assigned to the paper recommended **Accept (Poster)**.  You can find the other set of reviews, along with any follow up discussion with the authors here:
https://openreview.net/forum?id=tW7L9dKZ0OM